# T cell receptor β-chain-targeting chimeric antigen receptor T cells against T cell malignancies

**Fanlin Li[1,7], Huihui Zhang[1,2,7], Wanting Wang[1,3], Puyuan Yang[1], Yue Huang[1], Junshi Zhang[1], Yaping Yan[1], Yuan Wang[1], Xizhong Ding[1], Jie Liang[1], Xinyue Qi[1], Min Li[1], Ping Han[1], Xiaoqing Zhang[1], Xin Wang[4], Jiang Cao[5], Yang-Xin Fu ⓘ[6] & Xuanming Yang ⓘ[1,3] ✉**

The success of chimeric antigen receptor (CAR) T cells in treating B cell malignancies comes at the price of eradicating normal B cells. Even though T cell malignancies are aggressive and treatment options are limited, similar strategies for T cell malignancies are constrained by the severe immune suppression arising from bystander T cell aplasia. Here, we show the selective killing of malignant T cells without affecting normal T cell-mediated immune responses in vitro and in a mouse model of disseminated leukemia. Further, we develop a CAR construct that carries the single chain variable fragment of a subtype-specific antibody against the variable TCR β-chain region. We demonstrate that these anti-Vβ8 CAR-T cells are able to recognize and kill all Vβ8⁺ malignant T cells that arise from clonal expansion while sparing malignant or healthy Vβ8⁻ T cells, allowing sufficient T cell-mediated cellular immunity. In summary, we present a proof of concept for a selective CAR-T cell therapy to eradicate T cell malignancies while maintaining functional adaptive immunity, which opens the possibility for clinical development.

T cell malignancies comprise a heterogeneous group of disorders, each reflecting a clonal evolution of dysfunctional T cells at various developmental stages by distinct mechanisms[1,2]. T cell malignancies include T-cell acute lymphoblastic leukemia (T-ALL), cutaneous and peripheral T-cell lymphoma (CTCL and PTCL), and adult T-cell leukemia (ATL)[3]. Although the response rates of adult and pediatric T cell leukemia are up to 90% with intensifying chemotherapy, approximately 2/3 of these patients relapse[4,5]. Salvage chemotherapy with allogeneic stem cell transplantation can only cure ~3–15% of relapsed patients[6]. For CTCL and PTCL patients, the progression-free survival rate remains at 20–30%[7]. The median overall survival (OS) times of

patients with acute, lymphoma, chronic, and smoldering ATL subtypes were 8.3, 10.6, 31.5, and 55.0 months, respectively, and the 4-year OS rates were 11%, 16%, 36%, and 52%, respectively[8]. T cell malignancies involve complex oncogenic pathways, such as the cell cycle regulation pathway[9,10], the NOTCH signaling pathway[11,12], and the proliferative-cytokine pathway[13], which makes it challenging to develop targeted small molecule therapeutics. Therefore, novel targeted therapies for T cell malignancies, particularly for relapsed and refractory patients, are urgently needed.

Chimeric antigen receptor (CAR)-modified T cells have been developed by generating numerous tumor-targeting T cells. A CAR

[1]Sheng Yushou Center of Cell Biology and Immunology, School of Life Sciences and Biotechnology, Shanghai Jiao Tong University, Shanghai 200240, China. [2]Shanghai Lung Cancer Center, Shanghai Chest Hospital, Shanghai Jiao Tong University, West Huaihai Road 241, Shanghai 200030, China. [3]Joint International Research Laboratory of Metabolic & Developmental Sciences, Shanghai Jiao Tong University, Shanghai 200240, China. [4]Shanghai Longyao Biotechnology Limited, Shanghai 201203, China. [5]Department of Hematology, Affiliated Hospital of Xuzhou Medical University, Xuzhou 221002, China. [6]The Department of Pathology, University of Texas Southwestern Medical Center, Dallas, TX 75390, USA. [7]These authors contributed equally: Fanlin Li, Huihui Zhang. ✉e-mail: xuanmingyang@sjtu.edu.cn

consists of an extracellular antigen-binding domain, a hinge domain, a transmembrane domain, intracellular domains with a costimulatory domain (such as 4-1BB and CD28), and an activation domain from CD3ζ[14,15]. Despite the promising antitumor efficacy of CAR-T cells against B cell-derived leukemia, developing CAR-T cells against T cell malignancies has been hindered by several challenges, such as contaminating tumor cells in CAR-T cells, CAR-T cell fratricide, and T cell aplasia[3,16]. Due to the lack of tumor-specific surface antigens, most CAR-T cells target T cell-differentiation markers, such as CD3[17], CD4[18], and CD7[19,20]. Since normal T cells express the same differentiation markers, CAR-T cells indiscriminately kill other CAR-T cells and normal T cells, leading to CAR-T cell fratricide and T cell aplasia, respectively. Although CAR-T cell fratricide is avoidable by ablating the target gene in CAR-T cells with gene editing[17,20], this significantly increases CAR-T cell manufacturing difficulties.

CD19-targeting CAR-T cell therapy against B cell malignancies causes B cell aplasia, which can be overcome by periodic intravenous immunoglobulin (IVIG) infusion[21]. Unfortunately, T cell aplasia causes profound immunosuppression, which is likely associated with high morbidity and mortality rates during infection and cancer. It has been challenging to develop alternative options to overcome T cell aplasia. Selecting antigens with limited or no expression in normal T cell populations is key for resolving T cell fratricide and aplasia. CAR-T cells against CD30[22,23], CD37[24], T cell receptor beta constant 1 (TRBC1)[25], and CD1a[26] have shown promising antitumor effects and reduced T cell aplasia in preclinical models. During T cell development, T cell receptor diversity results from variable-diversity-joining (VDJ)-domain recombination, resulting in unique TCR Vβ and Vα features in specific T cell clones. The TCRβ locus is located on chromosome 7, with a cluster of 52 functional Vβ gene segments, a single D gene segment, thirteen J gene segments and two TRBC genes, and there are 24 families of Vβ gene segments in humans. Consequently, each TCR Vβ is used by only 0.58–10.84% of the TCR repertoire[27]. Usually, only one specific TCR, Vβ and Vα is used in T cell malignancies, due to T cell clonality, suggesting that malignancy-specific TCR Vβ or Vα usage could be targeted to discriminate malignant and normal T cells. However, this approach has not been applied to CAR-T cell design.

Here, we establish anti-Vβ8 and Vβ5 CAR-T cells that selectively kill a fraction of normal T cells and entire Vβ8+ or Vβ5+ T cell lymphoma or leukemia cells, but not other normal Vβ8− or Vβ5− T cells. Unlike nonselective approaches targeting entire T cell populations, Vβ8+ T cell-depleted mice show sufficient antibody responses and anti-tumor immunity. Our results provide an alternative strategy for CAR-T cell therapy against T cell malignancy with limited T cell fratricide and aplasia.

## Results

### T cells transduced to express TCR V domain-targeting CARs specifically eliminated the target-positive normal T cell populations in vitro

In contrast to polyclonal TCR usage in peripheral T cells (Supplementary Fig. 1a), T cell leukemia cell lines Jurkat (Vβ8+) and Molt-4 (mainly Vβ2+) demonstrated monoclonality (Supplementary Fig. 1b, c), consistent with previous findings[28]. These data collectively demonstrated the TCR Vβ monoclonality feature of T cell malignancies, which may serve as an immunotherapeutic target against T cell malignancies. Normal T cells are polyclonal in terms of TCR Vβ usage, and elimination of a fraction of selective TCR Vβ-positive T cells may not severely affect the integrity of the total T cell repertoire (Supplementary Fig. 1d).

Next, we investigated whether TCR Vβ subtype-specific antibodies could be used for CAR design using commercially available antibodies against Vβ8, Vβ5, and Vβ13. These antibodies specifically recognized a fraction of T cells from different donors (Supplementary Fig. 2). We used mass spectrometry to identify the amino acid sequences of these

antibodies and generated corresponding single-chain variable fragment (scFv)-based recombinant antibodies. These antibodies recognized a fraction of T cells, similar to the parental antibodies (Supplementary Fig. 3). We used these scFv sequences to generate 4-1BB- and CD3ζ-based second-generation CARs. After lentiviral transduction of activated T cells, the CARs were stably expressed on activated primary donor-derived T cells (Supplementary Fig. 4). We analyzed the percentage of Vβ-specific T cells as indirect evidence of Vβ-specific killing ability. Indeed, all the CAR-T cells eliminated CAR target-positive normal T cells (Fig. 1a), and no difference was found in total TCRαβ+ expression compared with mock-transduced T cells (Fig. 1b). However, the absence of selective TCR-Vβ-positive staining possibly reflected epitope blocking by the ligated anti-TCR Vβ CAR. Therefore, we analyzed TCR usage at the mRNA level by quantitative PCR. Consistent with the flow cytometric data, the mRNA expression data indicated that CARs targeted and eliminated the corresponding TCR Vβ population (Fig. 1c). After Vβ8-specific CAR-T cells were cocultured with freshly isolated T cells, we observed similar elimination of the freshly added Vβ8+ T cell population (Fig. 1d). The levels of the cytotoxicity-related cytokines interferon (IFN)-γ and tumor necrosis factor (TNF)-α, were increased (Fig. 1e). These data collectively demonstrated that TCR-Vβ-targeting CAR-T cells can discriminate target-positive and -negative T cells in the normal T cell repertoire.

### Vβ8-CAR-T cells specifically targeted Vβ8+ T cell leukemia cells in vitro

After demonstrating Vβ-CAR-T cell-mediated specific lysis of normal Vβ+ T cells, we asked whether these CAR-T cells could specifically recognize and lyse malignant T cells. Among the antibodies and malignant T cell lines we tested, Jurkat T cell leukemia cells and Vβ8-CAR-T cells were the only matched pair; thus, we used them for a proof-of-concept study. We generated and cocultured TCR Vβ8+ Lenti-X 293 cells with control T cells or Vβ8-CAR-T cells. Whereas the control T cells did not secrete IFN-γ and TNF-α in response to any target cells, Vβ8-CAR T cells specifically secreted IFN-γ and TNF-α only when incubated with Lenti-X 293-Vβ8+ cells but not Lenti-X 293-Vβ8− cells (Fig. 2a). We then cocultured control T cells or Vβ8-CAR-T cells with near-infrared fluorescent protein (IRFP)-expressing Jurkat-Vβ8+ and Jurkat-Vβ8− cells. When evaluating IRFP+ cells, control Vβ8+ and Vβ8− Jurkat T cells did not show differences in terms of cytotoxicity, whereas Vβ8-CAR-T cells specifically lysed Vβ8+ Jurkat cells (Fig. 2b). Consistently, when Jurkat-Vβ8+ cells and Vβ8-CAR-T cells were cocultured, Vβ8-CAR-T cells produced more effector molecules, including TNF-α, IFN-γ, and granzyme B (Fig. 2c, d), and Jurkat-Vβ8+ cells showed increased apoptosis based on elevated active caspase-3 expression (Fig. 2e). Therefore, Vβ8-CAR-T cells can specifically target Vβ8+ T cell leukemia cells in vitro.

### Vβ8-CAR-T cells were functional after repetitive antigen stimulation

Long-term proliferation and cytotoxic abilities are critical for CAR-T cell antitumor activity. To mimic the long-term exposure of CAR-T cells in a tumor-bearing host in vivo, we developed a repetitive antigen-stimulation protocol using irradiated Raji-Vβ8+ cells (Fig. 3a). Vβ8-CAR-T cells were stimulated four times every 6 days with irradiated Raji-Vβ8+ cells. The T cells were harvested 4 days after each stimulation and then incubated with live IRFP-Jurkat-Vβ8+ cells at various T cell:Jurkat cell ratios. We examined the effect of repetitive stimulation on CAR+ cells generated with T cells from multiple donors. Unlike other T cell-specific targets (i.e., CD3 or CD5) used for CAR constructs, these Vβ8-CAR-T cells did not show obvious CAR-T cell fratricide, and gene editing was not required for CAR-T cell expansion, which was demonstrated by an enriched CAR+ population and greater cell numbers after antigen-specific stimulation (Fig. 3b, c). We further

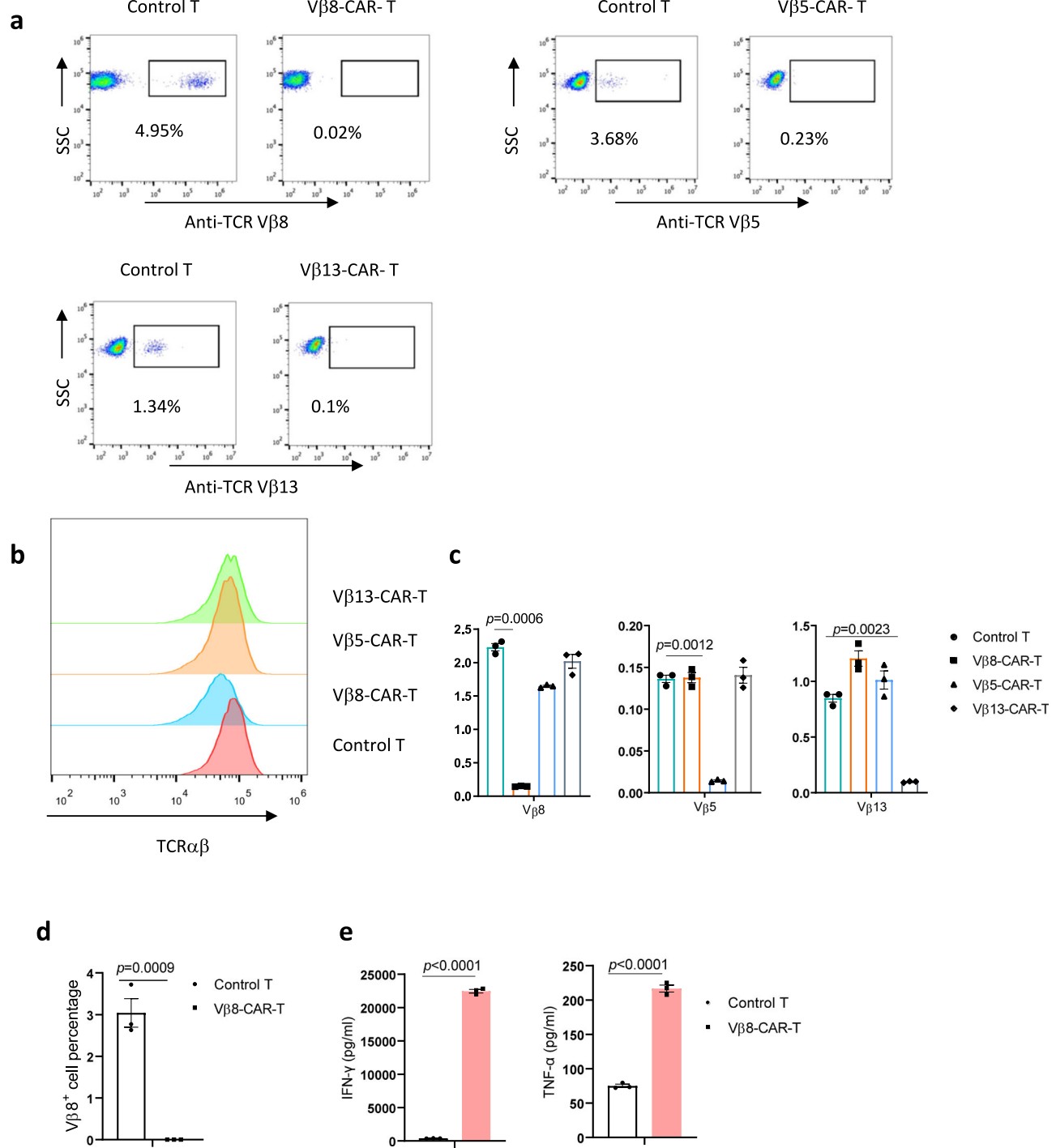

**Fig. 1 | Vβ-specific CAR-T cells specifically eliminated target-positive primary T cells. a–c** The expression levels of the indicated TCR Vβ subtypes (**a**, **c**) or total TCRαβ (**b**) in untransduced T cells and Vβ8-, Vβ13-, or Vβ5-specific CAR-T cells were analyzed by flow cytometry (**a**, **b**) or real-time PCR ($n = 3$ biologically independent samples/group) (**c**). **d**, **e** Untransduced T cells or Vβ8-CAR-T cells were cocultured with nonfractional normal T cells in triplicate wells at a CAR-T:normal T cell ratio of 10:1 for 24 h ($n = 3$ biologically independent samples/group). In the remaining Vβ8+ cells, cytotoxicity was determined by flow cytometry (**d**), and the indicated cytokines in the supernatants were analyzed by performing CBA assays (**e**). The data shown represent the mean ± standard error of the mean (SEM). Statistical significance was determined by two-sided unpaired t-tests. The normality of the data was confirmed by the Shapiro–Wilk test. Representative results of one of three replicate experiments are shown in **a–e**.

compared the cytotoxic ability of CAR-T cells after different antigen-stimulation periods. Vβ8-CAR-T cells killed significantly more IRFP-Jurkat Vβ8+ cells than control T cells at all tested activation stages (Fig. 3d). We also observed a trend toward slightly decreased cytotoxic ability after long-term stimulation, which was expected since T cell exhaustion and apoptosis occur under this condition. Since a subset of nonmalignant T cells will express a targeted Vβ8, this may induce CAR-T cell exhaustion during manufacture. We first observed that in different donor-derived Vβ8-CAR-T cells, the Vβ8+ cells were entirely eliminated 4 days after transduction (Supplementary Fig. 5). To explore whether a high percentage of Vβ8+ cells could affect CAR-T exhaustion, we established a range of Vβ8+ cells from 1 to 32% during

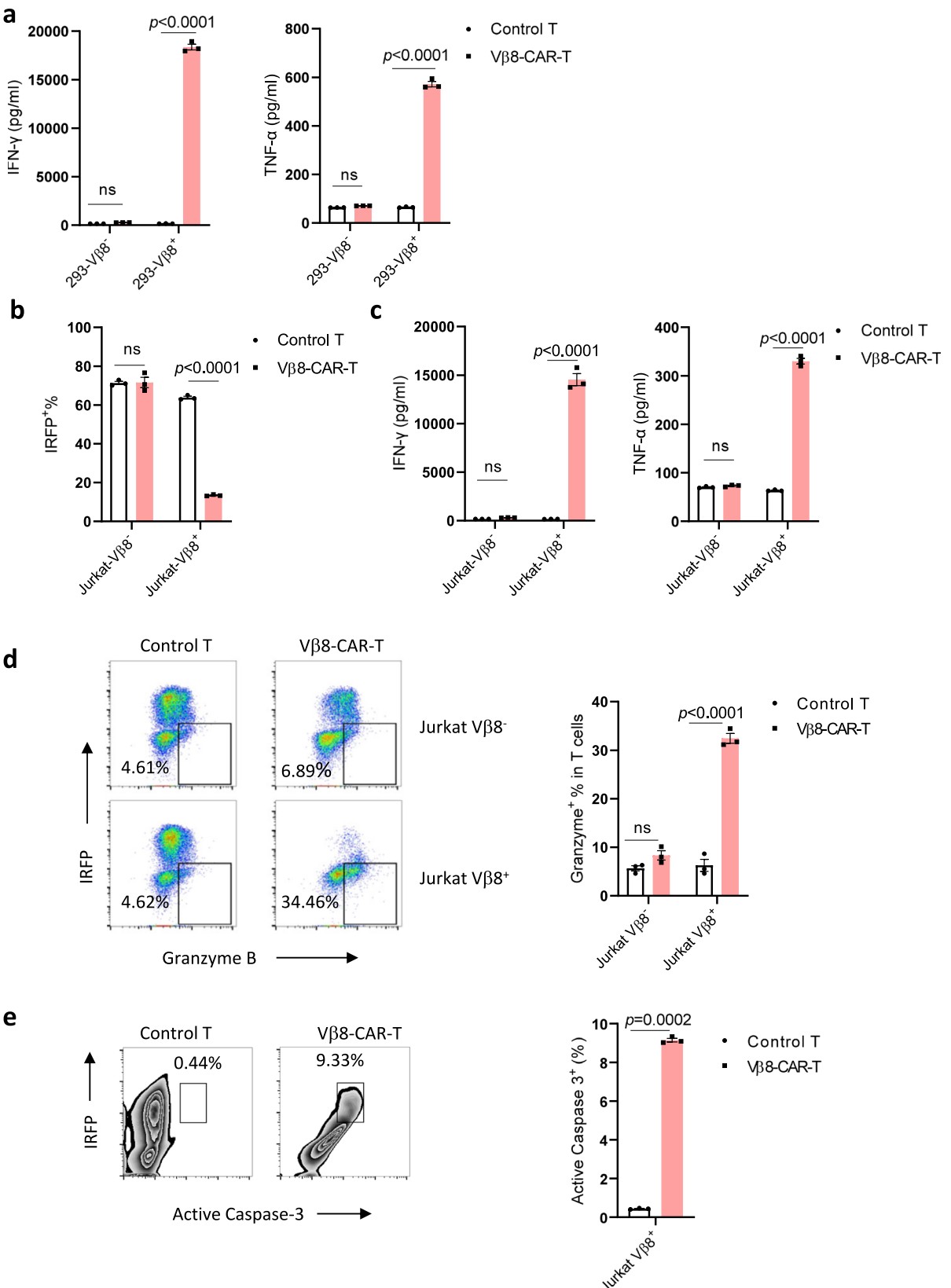

CAR-T cell culture by adding exogenous Jurkat cells. When the percentage of Vβ8+ cells was over 16%, more CAR-T cell exhaustion was induced, as indicated by increased PD-1 expression (Supplementary Fig. 6). This result suggested that it is critical to ensure that the percentage of Vβ8+ cells is below 16%. It would be better to include a negative selection process to exclude the Vβ8+ cell during Vβ8+-CAR-T cell manufacture. These data collectively demonstrated that Vβ8-CAR-T cells were capable of long-term proliferation, similar to other CAR-T cells, and their cytotoxic abilities were well maintained after long-term tumor antigen exposure.

**Fig. 2 | Vβ8-CAR-T cells specifically targeted Vβ8⁺ T cell leukemia cells in vitro.** **a** Untransduced T cells or Vβ8-CAR-T cells were cocultured with Lenti-X 293-Vβ8⁻ or Lenti-X 293-Vβ8⁺ cells in triplicate wells at a CAR-T:Lenti-X 293 cell ratio of 2:1 for 24 h (n = 3 biologically independent samples/group). The indicated cytokines in the supernatants were analyzed by performing a CBA assay. The data shown represent the mean ± SEM. Statistical significance was determined by two-sided, two-way ANOVA with Sidaks multiple comparisons test. **b**–**e** Untransduced T cells or Vβ8-CAR-T cells were cocultured with Jurkat-Vβ8⁻ or Jurkat-Vβ8⁺ cells in triplicate wells at a CAR-T: Jurkat cell ratio of 1:2 for 24 h (n = 3 biologically independent samples/group). Cytotoxicity was determined by analyzing the remaining IRFP⁺ cells with flow cytometry (**b**). The indicated cytokines in the supernatants were analyzed by performing CBA assays (**c**). Granzyme B (**d**) and active caspase-3 (**e**) were analyzed by flow cytometry. The data shown represent the mean ± SEM. Statistical significance was determined by two-sided unpaired t-tests. The normality of the data was confirmed by the Shapiro−Wilk test.

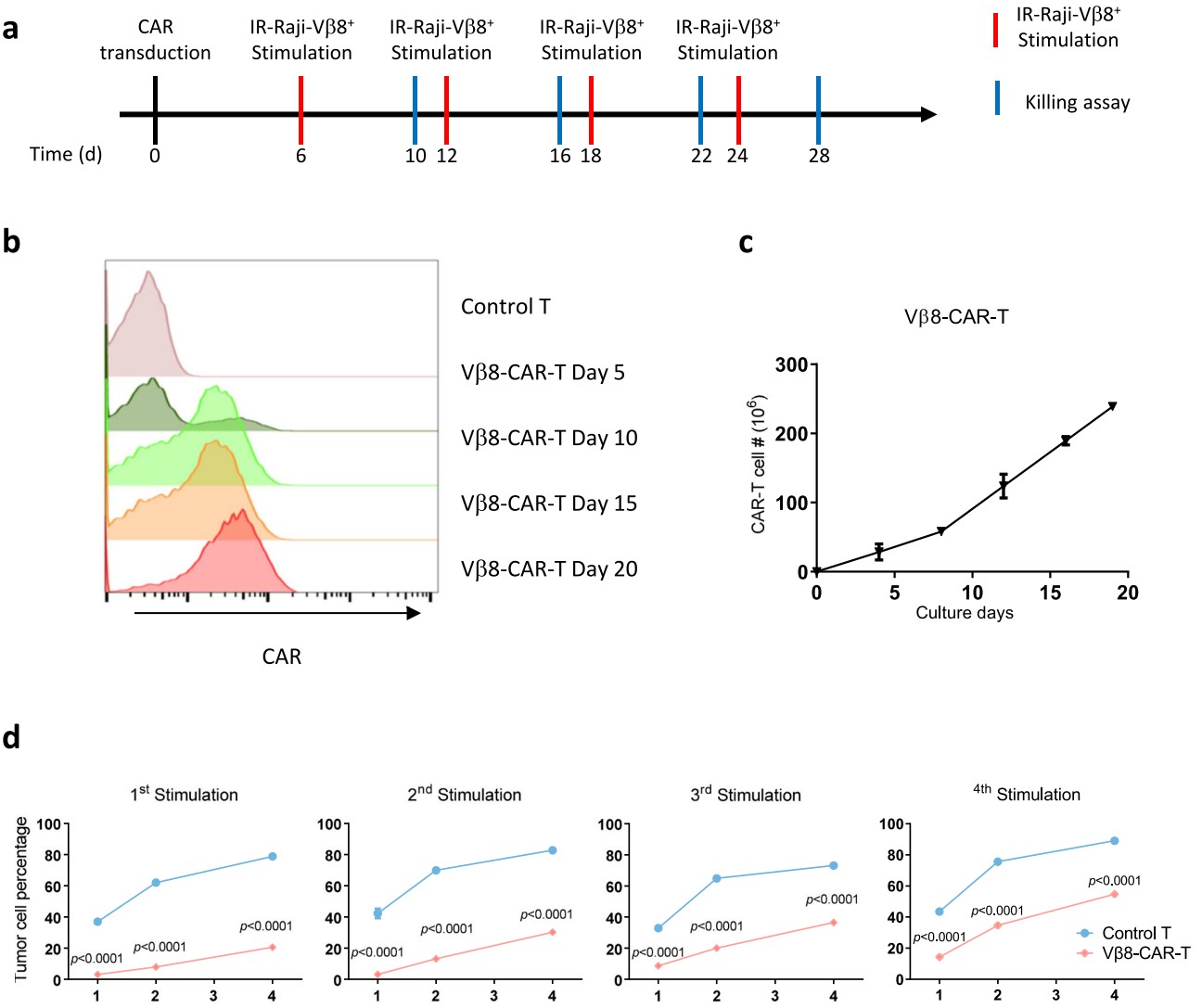

**Fig. 3 | Characterization of Vβ8-CAR-T cells after repetitive antigen exposure.** **a** Schematic representation of the CAR-T cell-characterization assay after long-term repetitive antigen stimulation. T cells were transduced with the indicated CAR lentivirus constructs and stimulated with irradiated Vβ8⁺ Raji cells once every 6 days. On Day 4 after each stimulation, CAR-T cells were studied by flow cytometry and cytotoxicity assays. **b** Vβ8-CAR-T cells were tested for CAR expression at the indicated time points by flow cytometry. **c** Vβ8-CAR-T cell growth curves during long-term culture (n = 4 biologically independent samples/group). **d** Vβ8-CAR-T cells generated as described in Panel A were cocultured with Jurkat cells in triplicate wells at different effector:target (T cell:Jurkat cell) ratios for 24 h (n = 3 biologically independent samples/group). Cytotoxicity was determined by analyzing the remaining IRFP⁺ cells with flow cytometry. The data shown represent the mean ± SEM from one of three replicate experiments. Statistical significance was determined by two-sided, two-way ANOVA with Sidaks multiple comparisons test.

## Vβ8-CAR-T cells were effective in a mouse model of T cell malignancy

To determine whether the specific cytotoxicity observed in culture with Vβ8-CAR-T cells correlates with antitumor efficacy in vivo, we established a Jurkat xenograft tumor model in NSG mice (Fig. 4a). Seven days after intravenous inoculation with Jurkat cells, the NSG mice received an intravenous injection of control T cells or Vβ8-CAR-T cells. Tumor cell burden in the peripheral blood, spleen, and bone marrow was compared at 27 days postinjection. The tumor burden in mice treated with Vβ8-CAR-T cells was significantly lower than that in mice treated with control T cells (Fig. 4b). The enhanced cytotoxicity correlated with the prolonged OS time of Vβ8-CAR-T cell-treated Jurkat tumor-bearing mice, compared to the control T cell-treated group (Fig. 4c). To further investigate dynamic tumor-burden changes after

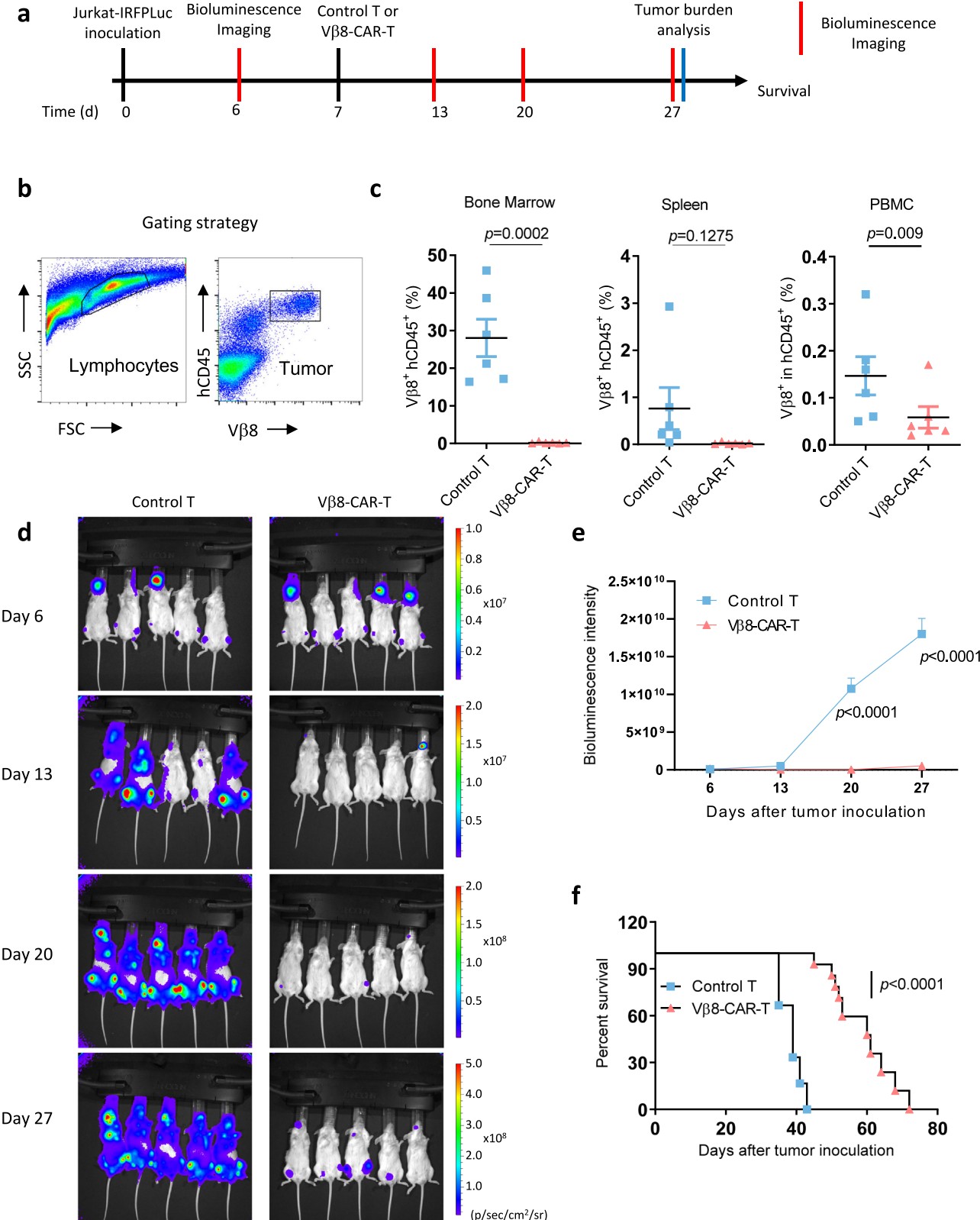

anti-Vβ8-CAR-T cell treatment, we engineered Jurkat cells expressing firefly luciferase and established xenograft models in NSG mice. Mice treated with Vβ8-CAR-T cells had a dramatic tumor burden reduction (determined by bioluminescence imaging) on Days 13, 20, and 27 after Jurkat cell inoculation (Fig. 4d, e), suggesting that the antitumor effect mediated by Vβ8-CAR-T cells was sustained.

**Vβ5-CAR-T cells showed similar antitumor activity as Vβ8-CAR-T cells in vitro and in vivo**

To determine whether this Vβ targeting strategy could be applied to different tumor models, we established a TCR Vβ5 targeting CAR-T cell and Vβ5 positive CCRF-CEM and Hut-78 T cell tumor cell lines (Supplementary Fig. 7). Similar to Vβ8 targeting CAR-T cells, we observed

**Fig. 4 | Vβ8-CAR-T cells were effective in vivo against T cell leukemia.**
**a** Schematic depiction of the Vβ8-CAR-T cell treatment schedule and sample analysis. **b, c** Female NSG mice aged 6–8 weeks (*n* = 6 mice/group) were intravenously inoculated with $2 \times 10^6$ Jurkat cells. Seven days later, tumor-bearing mice were treated with $1 \times 10^7$ untransduced T cells or Vβ8-CAR-T cells. Twenty days after treatment, bone marrow cells, spleen cells, and PBMCs were collected to analyze the Jurkat (hCD45[+]Vβ8[+]) tumor cell burden (**c**). The data shown represent the mean ± SEM (*n* = 6 mice/group). Statistical significance was determined by the two-sided Mann–Whitney U test. Representative results of one from two replicate experiments are shown. **d, e** Female NSG mice aged 6–8 weeks (*n* = 5 mice/group)

were intravenously inoculated with $2 \times 10^6$ Jurkat-IRFP-Luc tumor cells. Seven days later, tumor-bearing mice were treated with $1 \times 10^7$ untransduced or Vβ8-CAR-T cells. Bioluminescence images were acquired and analyzed at the indicated time points (**d, e**). Statistical significance was determined by two-sided, two-way ANOVA with Sidaks multiple comparisons test. Representative results of one from two replicate experiments are shown. **f** Kaplan–Meier OS curves are shown for the inoculated mice (*n* = 6 mice for control T group and *n* = 10 mice for Vβ8-CAR-T group). Statistical significance was determined by the two-sided Mantel–Cox test. Pooled data from two replicate experiments are shown.

Vβ5-positive normal T cell depletion during CAR-T cell manufacture (Fig. 5a). Furthermore, Vβ5 targeting CAR-T cells specifically lysed Vβ5-positive CCRF-CEM and Hut-78 leukemia cells in vitro (Fig. 5b). Consistent with the Vβ5-dependent cytotoxicity, Vβ5-CAR-T cells produced more IFNγ when cocultured with Vβ5-positive CCRF-CEM or Hut-78 tumor cells (Fig. 5c). We then evaluated the antitumor efficacy of Vβ5 CAR-T cells in vivo. CCRF-CEM-Vβ5[+]-bearing NSG mice were intravenously treated with control T cells or Vβ5-CAR-T cells. Vβ5-CAR-T cell treatment significantly reduced the tumor cell burden in the peripheral blood on Days 9 and 21 posttreatment (Fig. 5d). Consistent with the reduced tumor burden, the Vβ5-CAR-T cell-treated CCRF-CEM-Vβ5[+] tumor-bearing mice survived significantly longer than the control T cell-treated mice (Fig. 5e). These data demonstrated that TCR Vβ targeting CAR-T cells can be applicable to various types of T cell malignancies.

### Vβ8[+] cell-depleted mice showed sufficient cellular and humoral immune responses

Since each unique subtype of TCR Vβ–positive cells represents only a fraction of the normal T cell repertoire, we hypothesized that depleting these T cells may not induce severe immune suppression compared with T cell aplasia caused by CAR-T therapy targeting the whole population of T cells (such as CD5, CD7, and CD3). To test this hypothesis, we established a mouse model with selective TCR Vβ8[+] T cell depletion. A mouse Vβ8-specific antibody recognized approximately 21.5% of normal mouse T cells (Supplementary Fig. 8). We constructed mouse Vβ8-specific CAR-T cells, which specifically killed Vβ8[+] normal T cells and Vβ8[+] tumor cells (Supplementary Fig. 8, 9). Furthermore, mouse Vβ8-specific CAR-T cells promoted the survival of C1498-Vβ8[+]-bearing mice (Supplementary Fig. 10). To mimic long-term Vβ8[+] T cell depletion, we generated an anti-Vβ8 recombinant antibody using the same scFv. We also generated an anti-mouse CD3ε antibody as a control. Using these antibodies, we selectively deleted Vβ8[+] cells or the entire T cell population from mice (Fig. 6a, b). To test whether selective Vβ8[+] T cell depletion would affect antibody production after vaccination, we vaccinated these mice with DEC-OVA using anti-CD40 as an adjuvant[29]. Anti-DEC-OVA antibody production was reduced in anti-CD3ε Ab-treated mice, whereas anti-Vβ8 Ab-treated mice produced a similar level of antigen-specific antibodies as control mice (Fig. 6c). This anti-CD40-based, combined vaccination strategy also generated strong OVA-specific CD8[+] T cell responses[29]. Consistently, we detected approximately 8% of OT-I-specific T cells in the peripheral blood via pentamer staining. Vβ8[+] cell-depleted mice showed similar (~10%) OT-I-specific T cell responses, whereas OT-I specific T cell responses were completely abolished in anti-CD3ε antibody-treated mice (Fig. 6d and Supplementary Fig. 11).

We further tested whether this deficiency could affect antitumor immunity. We inoculated mice with B16-OVA tumor cells and compared the tumor growth curves. When all T cells were depleted with the anti-CD3ε antibody, the tumors grew faster than in immunocompetent wild-type (WT) mice. However, in Vβ8[+] T cell-depleted mice, the tumor-growth curve was similar to those of WT mice. These data collectively suggested that TCR Vβ8[+] T cell-depleted mice could mount sufficient

cellular and humoral immune responses, whereas mice with T cell aplasia were severely deficient in both arms of immunity.

## Discussion

Selective usage of the TCR V domain is widely recognized for its role in generating TCR diversity. We demonstrated that it is possible to discriminate TCR Vβ domains on normal and malignant T cells for CAR-T therapy. We generated TCR Vβ8-specific CAR-T cells and explored their potential for treating T cell malignancies (Fig. 7). Our results demonstrated TCR Vβ8-specific CAR-T cells selectively eliminated a fraction of Vβ8[+] normal T cells and malignant Vβ8[+] T cell leukemia cells. Compared with the other CAR-T strategies for T cell malignancies, this approach has several advantages: (i) it avoids CAR-T cell fratricide in the absence of gene editing; (ii) it maintains the relative T cell repertoire integrity to avoid T cell aplasia, and (iii) it is applicable for a subgroup of T cell malignancies independent of a complicated oncogenic pathway.

Due to the unselective elimination of normal T cells and malignant cells, pan-T cell antigen-targeting CAR-T cells, such as CD5, CD7, and CD3 T cells[17–20,30], will most likely be useful as a temporary tumor burden-control strategy instead of a permanent treatment option. Suicide strategies can be used to eliminate these CAR-T cells when the tumor-control goal is achieved[31,32], which will provide a therapeutic window for other treatments. To selectively eliminate T cell malignancy and maintain the integrity of T cell immunity, antigens with limited expression on T cells (e.g., CD30, CD37, TRBC1, and CD1a) were selected for CAR-T therapy[22,24–26]. CD37 is a member of the tetraspanin superfamily with expression limited to lymphoid tissues, particularly B cells. Although CD37 is predominantly being examined for targeting B cell malignancies, it can also be expressed in CTCLs and PTCLs. CD37 has potential for CAR therapy against T cell malignancies[24]. CD30, a member of the TNF-receptor family, is expressed in some activated lymphocytes around the follicular regions of lymphoid tissues, with strong expression in virtually all Hodgkin lymphoma cases and a subset of PTCLs and T-ALLs after chemotherapy[22]. Both targets have been tested both in preclinical models and in patients. However, only some patients with T cell malignancies benefit from these treatments, as most patients do not express either target. TRBC1-targeted CAR-T cells effectively controlled T cell malignancies in preclinical models[25]. The beta chain constant region of αβ T cells is encoded by either the TRBC1 or TRBC2 gene. The median percentage of TRBC1-positive normal T cells is approximately 35%. However, as malignant T cells develop from a single cell, the entire population of cancerous cells will be either TRBC1- or TRBC2-positive. Therefore, targeting TRBC1 can effectively eliminate TRBC1-positive tumor cells, with a loss of approximately 1/3 of the total T cell population. With the remaining 2/3 of the T cell repertoire, the host is expected to maintain the relative integrity of cellular immunity against tumors and infectious diseases, in contrast to T cell aplasia caused by pan-T-targeting CAR-T therapy. TRBC1-targeting CAR-T therapy is being evaluated in human clinical trials, which will provide the safety and efficacy profile in human patients. By targeting different TCR Vβ domains, our

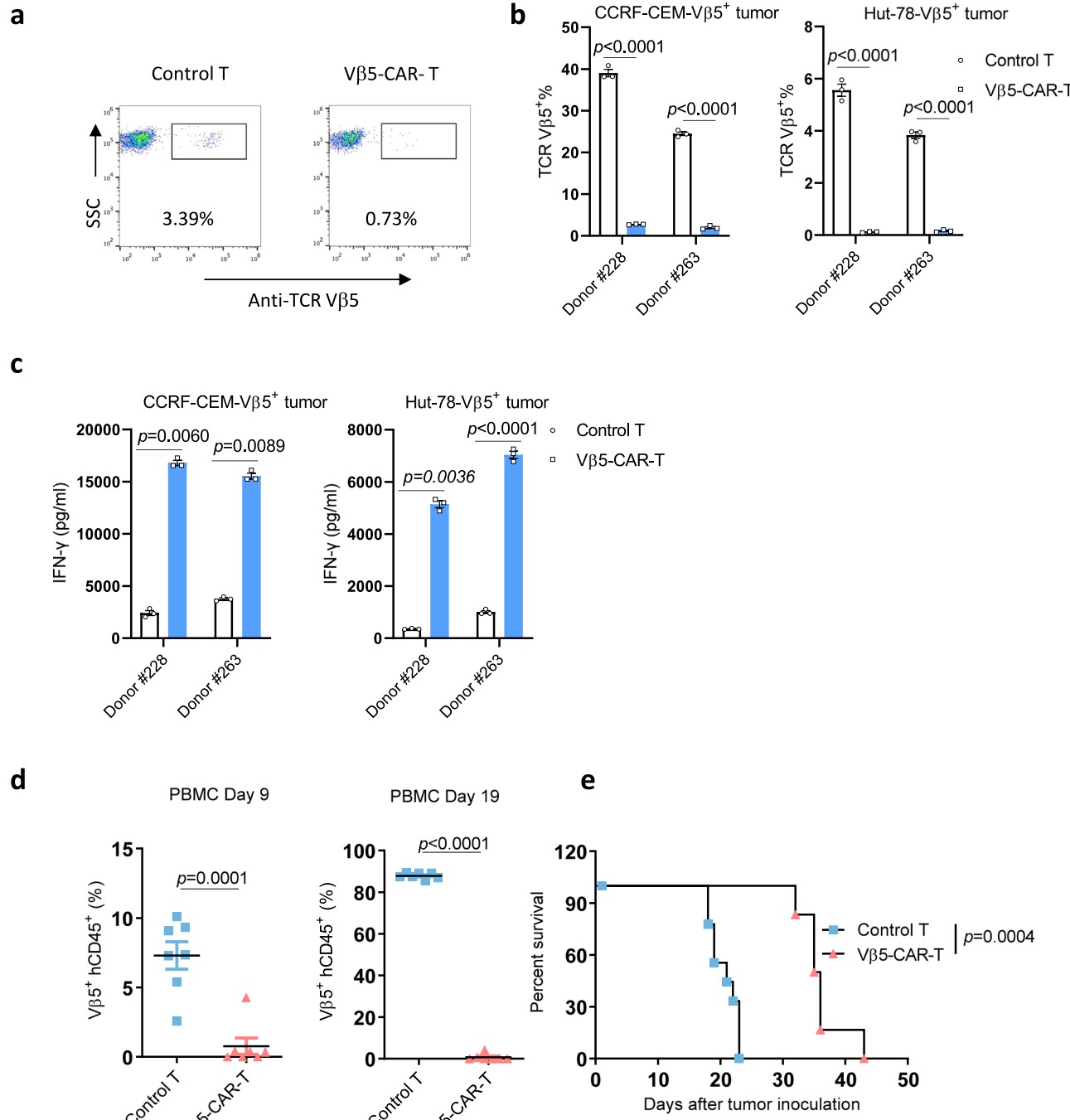

**Fig. 5 | Vβ5-CAR-T cells were effective in vitro and in vivo against T cell leukemia. a** The expression levels of the TCR Vβ5 subtypes in untransduced T cells or Vβ5-specific CAR-T cells were analyzed by flow cytometry. **b, c** Untransduced T cells or Vβ5-CAR-T cells were cocultured with CCRF-CEM-Vβ5⁺ or Hut-78-Vβ5⁺ cells in triplicate wells at a CAR-T:tumor cell ratio of 2:1 for 24 h (*n* = 3 biologically independent samples/group). The remaining Vβ5⁺ cells, was determined by flow cytometry (**b**), and the indicated cytokines in the supernatants were analyzed by performing CBA assays (**c**). **d** female NSG mice aged 6–8 weeks (*n* = 7 mice/group) were intravenously inoculated with 1 × 10⁶ CCRF-CEM-Vβ5⁺ cells. Seven days later, tumor-bearing mice were treated with 1 × 10⁷ untransduced T cells or Vβ5-CAR-T

cells. Nine and nineteen days after treatment, PBMCs were collected to analyze the CCRF-CEM-Vβ5⁺ (hCD45⁺Vβ5⁺) tumor cell burden. Statistical significance was determined by two-sided unpaired t tests. The normality of the data was confirmed by the Shapiro−Wilk test. Representative results of one from two replicate experiments are shown. The data shown represent the mean ± SEM (**b-d**). **e** Kaplan−Meier OS curves are shown for the inoculated mice (*n* = 9 mice for control T group and *n* = 6 mice for Vβ5-CAR-T group). Statistical significance was determined by the two-sided Mantel−Cox test. Representative results of one from two replicate experiments are shown.

approach can theoretically be used to treat any TCRαβ-positive T cell malignancy, compared with CD30- and CD37-targeted CAR-T therapies. Our TCR Vβ domain-specific CAR-T treatment eliminated < 11% of the normal T cell repertoire, which may provide close to intact cellular immunity, in contrast with pan-T-targeting

CAR-T therapies. The same strategy can also be used for Ab-based cancer therapy against T cell malignancies. During our manuscript submission, Pual et al. reported that TCR Vβ is a potential target for designing a bispecific T-cell engager against T-cell lymphoma[33]. Similar to targeting TRBC1, kappa and lambda light chains have

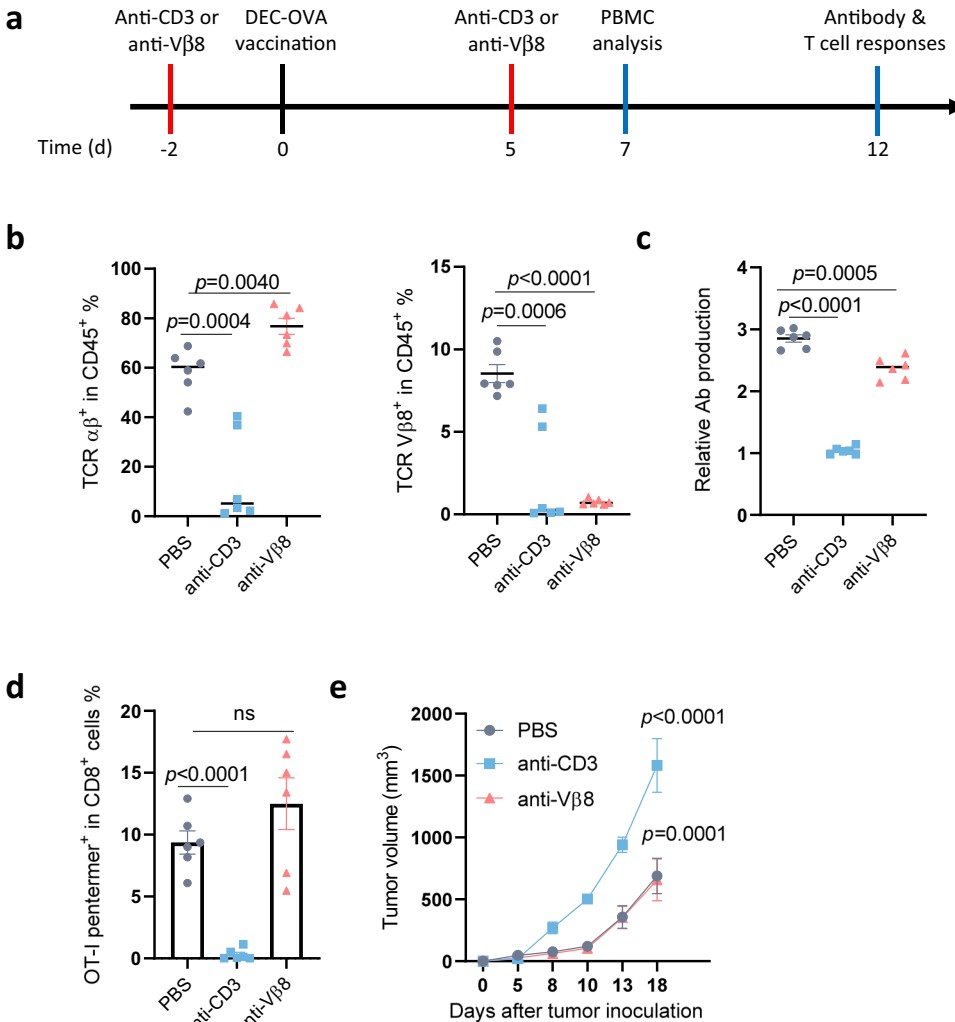

**Fig. 6 | Sufficient T cell-mediated immunity after TCR Vβ8+ cell depletion.**
**a** Schematic representation of the procedure used for analyzing cellular and humoral immune responses in Vβ8+ cell-depleted mice. **b**–**e** Female WT B6 mice aged 6–8 weeks (*n* = 6 mice/group) were intraperitoneally injected with 200 µg of anti-CD3 or anti-TCR Vβ8 antibodies or PBS on days −2 and 5. The mice were vaccinated with 10 µg of the DEC-OVA fusion protein and 100 µg of an anti-CD40 antibody as an adjuvant (**b**–**d**). On Day 7, peripheral blood TCRαβ+ or Vβ8+ cells were analyzed by flow cytometry (**b**). On Day 12, DEC-OVA-specific antibodies were analyzed by ELISA (**c**), and OT-I-specific CD8+ T cell responses in the peripheral

blood were analyzed by OT-I-pentamer staining (**d**). The data shown represent the mean ± SEM Statistical significance was determined by two-sided, one-way ANOVA with Dunnett's multiple comparisons correction. ns, not significant. Mice were prepared as described in Panel A and were subcutaneously inoculated with $5 \times 10^5$ B16-OVA cells (*n* = 6 mice/group) (**e**). Tumor sizes were measured and compared twice a week. Representative results of one from two replicate experiments are shown. The data shown represent the mean ± SEM Statistical significance was determined by two-sided, two-way ANOVA with Sidaks multiple comparisons test.

been used as targets of CAR-T therapy against B cell malignancies to avoid B cell aplasia[34,35], which only depletes kappa or lambda chain positive B cells. Considering the similar V-D-J recombination mechanism during B cell maturation and the monoclonal nature of B cell malignancies, it will be interesting to investigate whether the B cell variable domain could be used as a CAR-T cell therapy target. We hypothesize that this strategy will have a limited impact on the normal B cell repertoire, in contrast to B cell aplasia caused by CD19 targeting CAR-T cells.

T cell malignancies are highly heterogeneous in terms of oncogenic-driver pathways; thus, targeted therapy options vary with different subtypes. However, our approach revealed an oncogenic signaling pathway independent treatment option for different types of T cell malignancies. Although each type of Vβ- or Vα-specific CAR-T cell can only treat a fraction of T cell leukemia or lymphoma cells, they can target all TCRαβ+ tumor cells after TCR Vα or Vβ CAR-T cells have been developed for all subtypes.

The study limitations include a lack of clinical efficacy and safety evaluations. Although TCR Vβ8-specific CAR-T cells showed strong antitumor efficacy in vitro and in vivo, a clinical trial is necessary to validate these findings. Our data showed comparable T cell immunity in Vβ8+ T cell-depleted mice. However, these mice were housed under specific pathogen-free conditions, and whether they could mount sufficient immune responses against natural infections is unclear. Furthermore, certain antiviral immune responses show biased TCR Vβ usage[36,37]. When using TCR Vβs as CAR-T therapy targets, immune responses in the absence of T cell subtypes against corresponding infections should be investigated. Similarly, human invariant NKT and MALT predominantly use defined Vα24-Jα18/Vβ11 or Vα7.2-Jα33/Vβ2 or 13, respectively[38,39]. CAR-T-based targeting of the V domain used in invariant NKT and MALT should be fully explored before clinical development. Finally, TCRs are not absolutely required for some malignant T cells, and these cells may escape TCR Vβ-specific CAR-T therapy by TCR

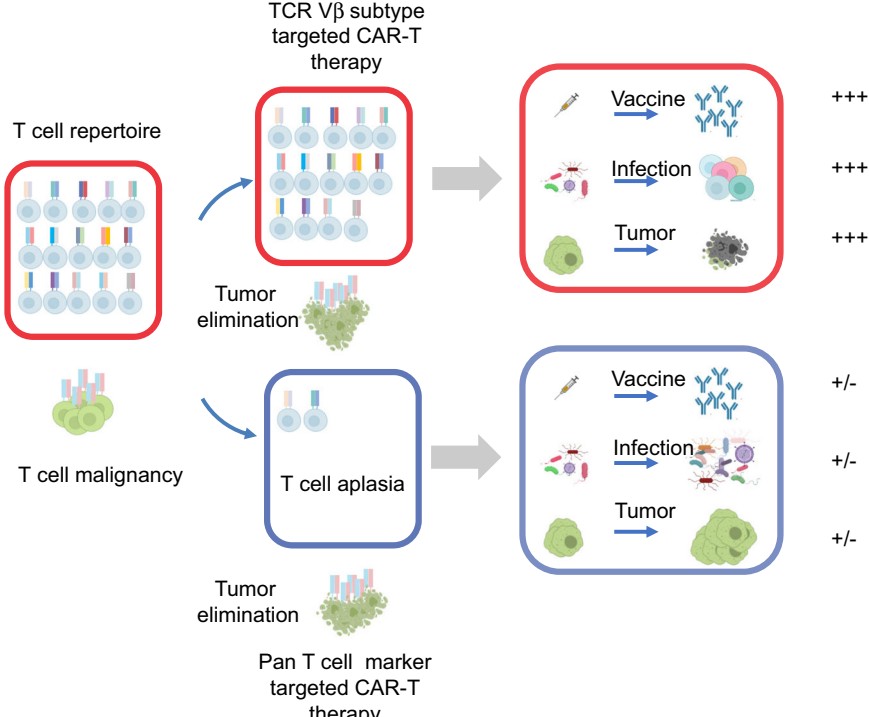

**Fig. 7 | Proposed model of how TCR Vβ-targeting CAR-T cell treatment eliminates T cell malignancies and maintains sufficient T cell-mediated immunity.** T cell malignancies are mostly monoclonal or oligoclonal. When pan T cell markers are used for CAR-T cell targets, CAR-T cells eliminate both malignant T cells and normal T cells. This strategy will cause T cell aplasia in the host and weak immune responses against vaccination, infection and tumor cells. When TCR Vβ subtypes are used for CAR-T cell targets, CAR-T cells eliminate malignant T cells and a small portion of normal T cells. This strategy will maintain sufficient T cell-mediated immunity during vaccination, infection and tumor control. The cartoons in this figure were created with BioRender (http://biorender.com/).

downregulation or loss of expression. It is also important to test whether TCR Vβ targeting CAR-T cells could be used with T cell malignancies to prevent tumor escape and relapse.

In summary, we developed a TCR Vβ-targeting CAR-T strategy for T cell malignancies, using the monoclonality of T cell malignancies to discriminate between normal T cells and tumor cells. Tumor cells were eliminated with a mild loss of the normal T cell repertoire, which avoids T cell aplasia-induced severe immune suppression. Our CAR design could be used to treat various T cell malignancies, independent of the intrinsic oncogenic driver-signal pathway, and it provides an alternative approach for treating T cell leukemia and lymphoma.

## Methods
### Mice
Female NOD-Prkdc^scid^IL2rγ^tm1^ mice (#NM-NSG-001) aged 6–8 weeks were purchased from Shanghai Model Organisms Center, Inc. Female C57BL/6 mice (# 213) aged 6–8 weeks were obtained from Beijing Vital River Laboratory Animal Technology Co., Ltd. (Beijing, China). All mice were maintained under SPF conditions. Mice were housed with a 12 h dark/light cycle, humidity of 40–70%, ambient temperature of 20–26 °C and enough supplement of food and water. The maximally permitted tumor burden is the diameter of 20 mm in any dimension, which was not exceeded during the experiments. Animal care and use were in accordance with institutional and NIH protocols and guidelines, and all studies were approved by the Animal Care and Use Committee of Shanghai Jiao Tong University.

### Cell lines and culture conditions
The Lenti-X 293 cell line was purchased from Clontech. The Phoenix-ECO cell line was purchased from ATCC. Jurkat, Molt-4, CCRF-CEM and Hut-78 cells were kindly provided by the Stem Cell Bank (Chinese Academy of Sciences). C1498 cells were kindly provided by Dr. Justin Kline (University of Chicago). B16-OVA cells were kindly provided by Dr. Hans Schreiber (University of Chicago). Human peripheral blood mononuclear cells (PBMCs) were purchased from Allcells Biotechnology, Limited (Shanghai, China) or provided by Shanghai Longyao Biotechnology Limited. Informed written consent was obtained from all study participants, and the protocol was approved by the Suqian Obstetrics and Gynecology Hospital Ethics Committee. To generate Lenti-X 293-Vβ8+ or Raji-Vβ8+ cells, Lenti-X 293 or Raji cells were transduced with a lentivirus that expressed human CD3 and Vβ8. After selection with blasticidin and puromycin, resistant cells were subcloned and identified by flow cytometric analysis. Jurkat cells were transduced with a near-infrared fluorescent protein (IRFP) and luciferase-expressing lentivirus. After puromycin selection, resistant cells were identified by flow cytometric analysis. To generate Jurkat-Vβ8− cells, *TRCA* and *TRCB* in Jurkat cells were edited by CRISPR/Cas9 with the following sgRNA: TCTCTCAGCTGGTACACGGC and CAAA-CACAGCGACCTCGGGT, respectively. TCRα and TCRβ-negative cells were sorted by flow cytometry. To generate CCRF-CEM-Vβ5+ and Hut-78-Vβ5+ cells, *TRCA* and *TRCB* were edited by CRISPR/Cas9 with the indicated sgRNAs. TCRα and TCRβ-negative cells were sorted by flow cytometry and infected with TCR Vα21 and TCR Vβ5 lentiviruses to generate stable cell lines. To generate C1498-Vβ8+ cells, C1498 cells were transduced with a lentivirus to express mouse CD3. After selection with puromycin, mouse Vβ8+ cells were sorted and subcloned. Lenti-X 293 cells were cultured in Dulbecco's modified Eagle's medium supplemented with 9% heat-inactivated fetal bovine serum (FBS; Gibco), 2 mM ʟ-glutamine, 55 μM β-mercaptoethanol, 100 U/ml penicillin, and 100 μg/ml streptomycin. Jurkat, Molt-4, CCRF-CEM, Hut-78 and human primary T cells were cultured in Roswell Park Memorial Institute 1640 medium containing 9% heat-inactivated FBS, 2 mM ʟ-glutamine, 55 μM β-mercaptoethanol, 100 U/ml penicillin, and 100 μg/

ml streptomycin. All cells were cultured in a humidified incubator at 37 °C with 5% CO$_2$.

## mAb mass spectrometry de novo sequencing analysis

The amino acid sequences of anti-human Vβ8, Vβ5, and Vβ13 antibodies were analyzed by mass spectrometry Do Novo sequencing package provided by Beijing Mingde Zhengkang Medical Research Co., Ltd. The amino acid sequence was then used as a template to synthesize cDNA.

## Production of recombinant antibody

The scFv cDNA-encoding antibodies against human TCR Vβ8, Vβ13, or Vβ5, mouse CD3ε (clone 145-2C11)[40] or mouse TCR Vβ8 (clone F23.1)[41] were synthesized by Genewiz (Suzhou, China) and cloned into the pCDH-EF1-MCS vector with a C-terminal human IgG1 Fc fusion. These plasmids containing anti-TCRVβ were transfected into Lenti-X 293 cells, and supernatants were collected for staining donor-derived PBMCs or purified by a Protein A column according to the manual (Repligen Corporation).

The DEC205-OVA recombinant protein was produced similarly to published methods[29]. The anti-DEC205 scFv[29] and OVA were separately cloned into a 'key-hole' Fc pair plasmids[42] and transfected into Lenti-X 293 cells. The supernatants were collected and purified by a Protein A column according to the manual (Repligen Corporation). Anti-CD40 antibody (clone FGK45, BE0016-2) was purchased from BioXcell.

## Generation of lentiviral and retroviral vectors

Lentiviral and retroviral plasmids containing CAR constructs were generated by standard molecular cloning methods. Briefly, a DNA fragment encoding a scFv recognizing human TCR Vβ8, Vβ13, or Vβ5; a CD8 hinge and transmembrane domain; and a 4-1BB and CD3ζ intracellular domain was generated by overlapping PCR and subcloned into a modified pCDH-EF1a vector. A DNA fragment encoding a scFv recognizing mouse TCR Vβ8, a CD8 hinge and transmembrane domain, and a 4-1BB and CD3ζ intracellular domain was generated by overlapping PCR and subcloned into a modified pMSGV vector. Complementary DNA (cDNA) for human TCR Vα17, Vβ8, or CD3 was PCR-amplified and subcloned into pCDH-EF1a-IRES-Puro to generate a Vβ8- or CD3-expressing plasmid.

## Lentivirus and retrovirus production

Lentiviruses were produced by transfecting Lenti-X 293 cells with a four-plasmid system. Supernatants containing lentivirus particles were collected at 48 and 72 h posttransfection and concentrated by ultracentrifugation. Retroviruses were produced by transfecting Phoenix-ECO cells with pCL-ECO and mouse TCR Vβ8 CAR retroviral plasmids. Viral titers in transduction units (TU) per ml were determined by flow cytometric analysis of transduced Lenti-X 293 cells or 3T3 cells.

## Transduction of human and mouse T cells

Human PBMCs from healthy donor umbilical cord blood were isolated with Human PBMC separation fluid (Tbd Science). PBMCs were cultured in RPMI 1640 complete medium supplemented with 50 IU/ml recombinant human (rh) IL-2 and 4 ng/ml rhIL-21. Plate-bound antibodies recognizing human CD3 and soluble antibodies recognizing CD28 were used to activate T cells in culture. Three days after activation, various amounts of CAR-containing lentiviruses were used to transduce T cells at a multiplicity of infection of 10. Mouse T cells were purified from C57BL/6 splenocytes using a Mouse CD3 isolation Kit (Stemcell). Purified mouse T cells were cultured with RPMI 1640 complete medium supplemented with 50 IU/ml recombinant human (rh) IL-2 and 4 ng/ml rhIL-21. Plate-bound antibodies recognizing mCD3 and soluble antibodies recognizing mCD28 were used to activate mouse T cells in culture. Two days after activation, activated T cells were infected with mouse TCR Vβ8 targeting retroviruses at an MOI of 2.

## Repetitive-stimulation assay

Lentivirus-transduced T cells were cultured with RPMI 1640 complete medium supplemented with 50 IU/ml rhIL-2 and 4 ng/ml rhIL-21. Irradiated (100 Gy) Raji-Vβ8$^+$ cells were used to stimulate these CAR-T cells at a ratio of 3:1 (T cells: Raji-Vβ8$^+$ cells), first on Day 6 postviral transduction and then once every 6 days during the entire culture period. The transduced T cells were maintained at a density of $1-2 \times 10^6$ cells/ml. The cells were monitored daily and fed according to cell counts every 1–2 days for 22–25 days before using in vitro or in vivo experiments. On Day 4 after stimulation with irradiated Raji-Vβ8$^+$ cells, the CAR-T cells were used for cytotoxicity assays and T cell-subtype analysis.

## Flow cytometry

For cell-surface staining, cells were incubated for 10 min in staining buffer (1× phosphate-buffered saline [PBS] with 1% FBS) with antibodies recognizing CD16 and CD32 (anti-FcγRII and FcγRIII, clone 2.4 G2, BE0307,1:250, BioXcell), or Human TruStain FcX (422302, 1:250, BioLegend) to block nonspecific Fc-mediated binding. The blocked samples were subsequently stained with conjugated antibodies. Intracellular staining was performed following fixation for 30 min at room temperature in 4% paraformaldehyde and permeabilization with 1× Perm/Wash buffer (eBioscience) for 60 min at 4 °C. The cells were incubated with the indicated antibodies diluted in 1× Perm/Wash buffer for 30 min at 4 °C. The cells were analyzed using a CytoFLEX S flow cytometer (Beckman Coulter), and the data were analyzed with FlowJo software v10 (Becton Dickinson).

The following fluorescently labeled monoclonal antibodies were used at the indicated dilutions: goat anti-mouse IgG, F(ab')2 fragment specific (115-605-006, 1:2500, Jackson ImmunoResearch), anti-human Fc (109-005-098, 1:2500, Jackson ImmunoResearch), anti-mouse Fab(115-005-071, 1:2500, Jackson ImmunoResearch) human Vβ8 (348104, clone JR2, 1:250, BioLegend), Vβ5 (349301, clone MH3-2, 1:250, BioLegend), Vβ13 (362402, clone H131, 1:250, BioLegend), active caspase-3 (560626, clone C92-605, 1:250, BD Biosciences), granzyme B (372206, clone QA16A02, 1:250, BioLegend), mouse CD45 (103126, clone 30-F11, 1:250, BioLegend), TCRVβ8.1 (118405, clone KJ16-133.18, 1:250, BioLegend), TCRβ (109205, clone H57-597, 1:250, BioLegend), and Kb-SIINFEKL-pentamer (F093-2A-PE) (1:50, PROIMMUNE).

## In vitro CAR-T cell cytotoxicity assay

CAR-T cells ($1 \times 10^5$) were cocultured with $1 \times 10^5$, $2 \times 10^5$, or $4 \times 10^5$ Jurkat tumor cells in a flat-bottom 96-well plate. Twenty-four hours after plating, the cells were harvested and analyzed by flow cytometry. IRFP expression was used to distinguish between CAR-T cells and tumor cells.

## Cytokine production analysis

The production of the cytokines IFN-γ and TNF-α in culture supernatants was quantified using Cytometric Bead Array (CBA) Kits (BD Bioscience) according to the manufacturer's instructions.

## Gene expression analysis by quantitative PCR

Total RNA from CAR-T cells was extracted with a Total RNA Kit (Omega Bio-Tek) and reverse transcribed into cDNA with ReverTra Ace reverse transcriptase (Toyobo). qPCR was performed in triplicate with SYBR Green Supermix (Toyobo). Relative mRNA expression levels were calculated by the 2-ΔΔCt method. TCR Vβ-specific primers were synthesized according to a previous publication[43].

## Animal models

To evaluate the antitumor activity of Vβ8-CAR-T cells against T cell leukemia, female NOD-Prkdc$^{scid}$IL2ry$^{tm1}$ mice aged 6–8 weeks were inoculated intravenously with $2 \times 10^6$ Jurkat cells. Approximately 7 days later, $1 \times 10^7$ untransduced T cells or Vβ8-CAR-T cells were infused intravenously, and animal survival was monitored over time. To evaluate in vivo tumor burdens, we harvested peripheral blood, spleens, and bone marrow from femurs and tibias, after which single-cell suspensions were prepared for flow cytometric analysis. The spleens were manually homogenized in PBS and passed through a 70 μm cell strainer (BD Biosciences). To obtain cells from the bone marrow, the femurs and tibias were flushed with PBS using a 1 ml syringe (KDL). ACK lysis buffer (BD Biosciences) was used to lyse red blood cells in all samples.

To evaluate the antitumor activity of Vβ8-CAR-T cells against T cell leukemia, female NOD-Prkdc$^{scid}$IL2ry$^{tm1}$ mice aged 6–8 weeks were inoculated intravenously with $2 \times 10^6$ firefly luciferase-expressing Jurkat cells. Approximately 7 days later, $1 \times 10^7$ untransduced T cells or Vβ8-CAR-T cells were infused intravenously, and tumor burden was analyzed by measuring luciferase activity with an IVIS Lumina XR Real-Time Bioluminescence Imaging System (Caliper Life Sciences).

Female C57BL/6 mice aged 6–8 weeks were intraperitoneally injected with 200 μg of anti-CD3 or anti-TCR Vβ8 antibodies, or PBS on days −2 and 5. Then the mice were used for vaccination or tumor challenge. For vaccination, mice were intraperitoneally injected with 10 μg of the DEC-OVA fusion protein and 100 μg of an anti-CD40 antibody as an adjuvant. On Day 12, DEC-OVA-specific antibodies were analyzed by ELISA. For tumor challenge, mice were subcutaneously inoculated with $5 \times 10^5$ B16-OVA cells on day 0. Tumor sizes were measured along three orthogonal axes (a, b, and c) and calculated as tumor volume = abc/2.

## Statistical analysis

Statistical analyses were performed using GraphPad Prism 8 (Graph-Pad Software, Inc.). Statistical significance was determined by an unpaired two-sided Student's t-test, Mann–Whitney U test, two-sided, one-way analysis of variance (ANOVA) with Dunnett's multiple comparisons correction or two-sided, two-way ANOVA with Sidaks multiple comparisons test. When an unpaired t-test was applied, the normality of the data was confirmed by the Shapiro–Wilk test with a P-value of > 0.05. A two-sided log-rank test was applied to assess mouse survival. Where indicated, $*P < 0.05$, $**P < 0.01$, $***P < 0.001$, and $****P < 0.0001$ were considered to reflect statistically significant differences.

## Reporting summary

Further information on research design is available in the Nature Research Reporting Summary linked to this article.

# Data availability

The data generated in this study are provided in the main text or Supplementary Information/Source Data file. Source data are provided with this paper.

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

## Acknowledgements
We thank Dr. Haixia Jiang from the Core Facility and Technical Service Center for SLSB, School of Life Sciences & Biotechnology, Shanghai Jiao Tong University for her help with cell sorting. X.Y. was supported by the National Natural Science Foundation of China (81971467), The National Key Research and Development Program of China (2016YFC1303400) and Shanghai Jiao Tong University Scientific and Technological Innovation Funds (2019QYA11). F.L. was supported by China Postdoctoral Science Foundation (2022M712073). The cartoons in Fig. 7 and supplementary Figure 1d were created with BioRender.com.

## Author contributions
X.Y., Y.-X.F., F.L., and H.Z. designed the overall project. F.L., H.Z., P.Y., J.L., W.W., Y.H., X.Q., J.Z., Y.Y., Y.W., M.L., P.H., X.Z. X.D., J.C., X.W., and X.Y. performed the experiments. F.L., H.Z., Y.-X.F., and X.Y. analyzed the results and wrote the manuscript.

## Competing interests
X.W. is an employee of Shanghai Longyao Biotechnology Limited. X.Y. is a consultant of Shanghai Longyao Biotechnology Limited. All other authors declare that they have no competing interests.
