## [Peer Review File · Nature Communications]

T cell receptor β -chain-targeting chimeric antigen receptor T cells against T cell malignanciesREVIEWER COMMENTS

Reviewer #1 (Remarks to the Author):

In their paper Li et al report first attempt to target the TRBV region for CAR-T cells therapy in T cell neoplasms. The proposed approach is novel and has the potential to be used in clinical trials. The results are well documented and the article is clearly written. There are no major issues concerning the merits of the paper.

Suggestions:

1. The first part of the results "T cell malignancy-derived cell lines and primary T cell cancers were clonally positive for a unique TCR VB is generally known and can be omitted.
2. The comparison of CAR-T cell effectiveness in B- and T-cell malignancies (leading to T cell aplasia by targeting the same surface antigens in normal and malignant T cells) has been described in the introduction, and doesn't have to be repeated in the discussion.

Minor points:

1. The number of mice used for experiments should be given.
2. In figure 5D the VB8 and the control CAR-T mice should be indicated.
3. The abbreviation iRFP (Infrared fluorescent protein) should be expanded at first use.
4. Page 4: "a single C gene" should read "two TRBC genes"
5. Jurkat is a T cell leukemia, not lymphoma cell line. Please correct throughout the entire manuscript.

Reviewer #2 (Remarks to the Author):

The authors present a proof-of-principle study for the feasibility of treating T-cell malignancies using CAR-T cells expressing a V β -family-specific antibody. The report is well-written and the topic very interesting. I have a few minor only point:

1. Discussion: "Different from solid tumors, both B and T cell malignancies do not form an immune-suppressive tumor microenvironment, which may limit the anti-tumor efficacy of CAR-T cells." -> this is not true, see for example PMID 21270444.
2. There are some instances, where the authors repeat themselves, e.g. the problems of invariant receptor targeting are mentioned both in the Introduction and in the Discussion, similar applies to the CD19-targeting for B-cell malignancige, to the statement that "each TCR V β is used by only 0.58–10.84% of the TCR repertoire" and some other points. The authors could shorten a bit the manuscript by reducing these redundancies

Reviewer #3 (Remarks to the Author):

Li et al develop TCR Vbeta subtype specific CAR T cells to more selectively target T cell malignancies, a major challenge given that most T cell malignancy associated antigens are expressed on a broad population of T cells.

Comments

- 1) Donor to donor variability is well-recognized in the CAR T cell field. The experiments describe replicates but it is not clear how many donor were utilized. This may be particularly important when targeting a V beta that would be variably represented across donors.
- 2) As noted by the authors and published by others, the presence of antigen during CAR T cell manufacturing can results in dysfunctional cells. Since a subset of non-malignant T cells will express a targeted V beta, this should be explored. For example, how many of a given V beta expressing T cells is necessary to induce exhaustion?
- 3) The authors use of a single tumor model to validate the approach. Additional models would strengthen the conclusions. It appears that the authors did generate additional V beta CARs that could be used.

REVIEWER COMMENTS

Reviewer #1 (Remarks to the Author):

In their paper Li et al report first attempt to target the TRBV region for CAR-T cells therapy in T cell neoplasms. The proposed approach is novel and has the potential to be used in clinical trials. The results are well documented and the article is clearly written. There are no major issues concerning the merits of the paper.

Suggestions:

1. The first part of the results “T cell malignancy-derived cell lines and primary T cell cancers were clonally positive for a unique TCR VB is generally known and can be omitted.

Reply: We thank the reviewer for this suggestion. We agree with the reviewer that for the researchers in T cell malignancy field, this information is known. We have moved this part to the supplemental Figure 1 to inform these researchers in other fields, which can help them to understand this CAR-T design concept.

2. The comparison of CAR-T cell effectiveness in B- and T-cell malignancies (leading to T cell aplasia by targeting the same surface antigens in normal and malignant T cells) has been described in the introduction, and doesn't have to be repeated in the discussion.

Reply: We thank the reviewer for the suggestion. We have removed the comparison part in discussion section to improve the readability accordingly.

Minor points:

1. The number of mice used for experiments should be given.
2. In figure 5D the VB8 and the control CAR-T mice should be indicated.
3. The abbreviation iRFP (Infrared fluorescent protein) should be expanded at first use.
4. Page 4: “a single C gene” should read “two TRBC genes”
5. Jurkat is a T cell leukemia, not lymphoma cell line. Please correct throughout the entire manuscript.

Reply: We thank the reviewer for the careful reading of our manuscript and providing these important revision suggestions.

1. We have included the numbers of mice in the revised figure legends as suggested.

2. We have pointed the V β 8 and control CAR-T treated mice with different legends in revised figure 4D (previous figure 5D) as suggested.

3. We have spelled out iRFP at its first use according the reviewer's suggestion.

4. We have revised it to "two TRBC genes" as suggested.

5. We have corrected Jurkat as a T cell leukemia cell line. We thank the reviewer for the valuable suggestion to improve the quality of our manuscript.

Reviewer #2 (Remarks to the Author):

The authors present a proof-of-principle study for the feasibility of treating T-cell malignancies using CAR-T cells expressing a V β -family-specific antibody. The report is well-written and the topic very interesting. I have a few minor only point:

1. Discussion: "Different from solid tumors, both B and T cell malignancies do not form an immune-suppressive tumor microenvironment, which may limit the anti-tumor efficacy of CAR-T cells." -> this is not true, see for example PMID 21270444.

Reply We thank the reviewer for the correction and providing the critical reference. We have removed this statement in our revised manuscript.

2. There are some instances, where the authors repeat themselves, e.g. the problems of invariant receptor targeting are mentioned both in the Introduction and in the Discussion, similar applies to the CD19-targeting for B-cell malignance, to the statement that "each TCR V β is used by only 0.58–10.84% of the TCR repertoire" and some other points. The authors could shorten a bit the manuscript by reducing these redundancies

Reply We thank the reviewer for careful reading and the valuable suggestion. We have removed "each TCR V β is used by only 0.58–10.84% of the TCR repertoire" and "The comparison of CAR-T cell effectiveness in B- and T-cell malignancies" to avoid repeating and to improve the readability of our manuscript.

Reviewer #3 (Remarks to the Author):

Li et al develop TCR Vbeta subtype specific CAR T cells to more selectively target T cell malignancies, a major challenge given that most T cell malignancy associated antigens are expressed on a broad population of T cells.

Comments

1) Donor to donor variability is well-recognized in the CAR T cell field. The experiments describe replicates but it is not clear how many donor were utilized. This may be particularly important when targeting a V beta that would be variably represented across donors.

Reply: We thank the reviewer for this comment. We agree with the reviewer that the CAR-T cell function varies from donor to donor. The efficacy of the TCR V β 8 targeting have been tested from 6 donors derived PBMC. In these donors, the percentage of V β 8⁺ ranges from 2.99-5.06%. We all generated V β 8 CAR-T cells successfully and demonstrated similar anti-TCR V β 8 activity in vitro. These data have been included in our revised supplementary figure 5.

2) As noted by the authors and published by others, the presence of antigen during CAR T cell manufacturing can results in dysfunctional cells. Since a subset of non-malignant T cells will express a targeted V beta, this should be explored. For example, how many of a given V beta expressing T cells is necessary to induce exhaustion?

Reply: We thank the reviewer for this comment. This point is critical for the successful CAR-T cell manufacture. To this purpose, we adjusted the percentage of V β 8⁺ cell from 1% to 32% during CAR-T cell manufacture by adding exogenous V β 8⁺ Jurkat cells. From our observation, when the percentage of V β 8⁺ cells is over 16 %, it will induce more CAR-T cell exhaustion as indicated by more PD-1 expression. It suggested that it is better to ensure the V beta 8⁺ cell to below 16% in CAR-T cell preparation. These data have been included in our revised supplementary figure 6. However, for clinical manufacture of CAR-T cells, it is better to include a negative selection process to enrich non-malignancy T cells by anti-TCR V β beads, which can help to improve the "purity" of CAR-T cells.

3) The authors use of a single tumor model to validate the approach. Additional models would strengthen the conclusions. It appears that the authors did generate additional V beta CARs that could be used.

Reply: We thank the reviewer for this comment. We agree with the reviewer it is important to demonstrate the efficacy in different tumor models. Our current available CAR-T cells (V β 5, V β 13 and V β 8) are not matching the T cell malignancy cell lines we can obtain (Jurkat, Molt-4, CCRF-CEM and Hut-78). To overcome these limitations, we have generated two additional V β 5⁺ CCRF-CEM and Hut-78 cells by knocking out the endogenous TCR α and TCR β by CRISPR/Cas9-based gene editing, then expressing TCR V α 21 and TCR V β 5. The V β 5 targeting CAR-T cells showed potent activity against CCRF-CEM-V β 5 and Hut-78-V β 5 cells in vitro. It also reduced the tumor burden and prolonged the survival of CCRF-CEM-V β 5 bearing NSG mice in vivo. These data have been included in our revised figure 5.

REVIEWERS' COMMENTS

Reviewer #1 (Remarks to the Author):

The author provided detailed answers to the questions posed. After a thorough revision, the paper is now suitable for publication.

Reviewer #2 (Remarks to the Author):

The authors have responded to the comments adequately

Reviewer #3 (Remarks to the Author):

No additional comments

REVIEWERS' COMMENTS

Reviewer #1 (Remarks to the Author):

The author provided detailed answers to the questions posed. After a thorough revision, the paper is now suitable for publication.

Reply:

We thank the reviewer for the comments and supporting on our manuscript.

Reviewer #2 (Remarks to the Author):

The authors have responded to the comments adequately

Reply:

We thank the reviewer for the comments and supporting on our manuscript.

Reviewer #3 (Remarks to the Author):

No additional comments

Reply:

We thank the reviewer for the comments and supporting on our manuscript.